# Does Brand Truth-Telling Yield Customer Participation? The Interaction Effects of CSR Strategy and Transparency Signaling

**DOI:** 10.3390/bs12120514

**Published:** 2022-12-15

**Authors:** Weiping Yu, Jun Zhou, Mingli He, Dongyang Si

**Affiliations:** Business School, Sichuan University, Chengdu 610064, China

**Keywords:** corporate social responsibility, CSR strategy, brand environmental responsibility (BER), BER participation, customer trust, customer–brand identification (CBI)

## Abstract

Customer participation in brand environmental responsibility is necessary for enterprises and consumers to co-create value. However, it is not yet clear why some corporate social responsibility (CSR) communications are more effective in attracting higher customer participation in a digitally transparent environment. Based on signal theory and social identity theory, this study examines the impact of the interactive effect of CSR strategy (proactive vs. reactive) and transparency signals (high vs. low) on customer trust (perceived integrity and perceived competence), customer–brand identification, and participation intention in brand environmental responsibility. We conduct a 2 × 2 study with 140 respondents. The findings reveal a significant interaction effect of CSR strategy and transparency signals on perceived integrity, perceived competence, and participation intention in brand environmental responsibility. Mediation analysis reveals that the impact of CSR strategy on participation intention is serially mediated via perceived trust and customer–brand identification and varies across different transparency levels.

## 1. Introduction

Ongoing climate change and environmental deterioration are serious threats to ecological and socioeconomic sustainability. Green sustainability goals have been incorporated into brand management and corporate social responsibility (CSR) management [1]. Companies should adopt environmental responsibility strategies and action plans and implement corresponding green branding [2,3]. Branding is an essential activity for companies to gain understanding, recognition, and support through internal and external communication [4]. Brands must now focus on green sustainability, establish long-term values, and build a brand image that meets public needs around stakeholder demands [5,6], and pay more attention to consumer demand for environmentally friendly green, and low-carbon products and services [7,8]. Corporate environmental responsibility forms a closed-loop from goal vision, strategic planning, measurement actions, information disclosure, and stakeholder communication to brand building based on green low-carbon management [9], and requires the participation of consumers to cocreate sustainable development value [10]. All efforts might be in vain if consumers do not identify with the brand’s environmental responsibility efforts, do not support corporate environmental responsibility initiatives, or even doubt corporate motives [11,12]. Therefore, it is vital to ensure the effectiveness of brand environmental responsibility (BER) communication and to encourage consumers to identify with and participate in the brand.

The signaling theory [13,14] assumes that enterprises use CSR to convey positive attributes (e.g., credible) to consumers, who in turn respond to these signals with positive attitudes and behaviors (e.g., identification and participation intention) [15]. According to the signaling theory, we also hold that different levels of transparency may change the effectiveness of CSR because a high transparency signal enables consumers to have a stronger positive perception of the enterprise so as to quickly capture the positive image of the enterprise. Social media makes information communication faster, easier, and more transparent [16]. Communicating through social media also maximizes management performance [17,18,19]. Increased information transparency through social media can strengthen the relationship between BER and customer participation [19]. Research on accounting [20,21], business management [22,23], and product services [24,25] has confirmed the importance of transparency signals. Despite growing attention to the importance of transparency in the commercial world, the current focus on transparency is mainly related to stock market transactions, financial disclosure, and CSR report disclosure. Less is known about the impact of transparency and CSR interactions on customer decision-making. It is unclear whether transparency signals and CSR strategies together influence the link between customer attitudes toward brands and their participation in environmental activities. Therefore, the first research question of this paper is to explore the impact of the interaction between CSR strategy and transparency signaling on consumers’ BER participation intention.

Brands’ proactive, authentic, and transparent environmental responsibility is expected to build stronger relationships between customers and the public [26], and customers will admire brands implementing environmentally responsible practices [27]. When customers make consumption decisions, they associate relevant brands with environmental protection and health, and they trust and identify more with brands that are more socially responsible. Moreover, according to the social identity theory (SIT) [28,29], consumers will classify themselves as group members through self-categorization [30,31] and show consistent cognition, values or behavior norms with them (e.g., being socially responsible, supporting environmental protection) [32,33]. From the perspective of SIT, we hold that social identity reflects the consistency between consumers’ expectations and perceptions of CSR strategy and helps better understand consumers’ BER participation intention and the process and results of responding to CSR strategy. However, the internal mechanism of customer participation under the interaction of CSR and transparency still lacks exploration. It is worth exploring whether consumers have a good psychological perception of CSR strategy and agree with the enterprise’s action. Therefore, the second research question of this paper is to verify whether customer trust and customer-brand identification have mediating effects.

This study uses signaling theory and social identity theory to discover the effects of corporate environmental responsibility strategies (proactive vs. reactive) and transparency signals (high vs. low) on customer trust (integrity and competence), customer–brand identification (CBI), and customer participation intentions. This study reveals the influence of brand social responsibility communication on consumer response behavior in this digitized and transparent environment. This result reveals why customers participate in environmental activities and contribute to environmental protection. This study offers an effective solution for selecting transparency signals and communication attribution in CSR strategies. It provides a strategic reference for building customer trust and brand identity and boosting interaction between brands and consumers.

## 2. Conceptual Background

### 2.1. Environmental Responsibility in the Food Service Industry

The European Commission defined corporate social responsibility as the voluntary incorporation of social and environmental issues into business operations and interaction with stakeholders. Corporate environmental responsibility improves employee and customer satisfaction and enhances corporate financial performance while integrating economic, social, and environmental benefits to advance management activities related to corporate environmental responsibility and promote sustainable development. As sustainable supply chain-related activities have developed, so the application of sustainable supply chain management has been explored [34,35]. Environmental responsibility practices include sustainable consumption of products and natural resources, recycling, reuse, waste disposal, environmental information disclosure, and environmental governance [9,26,36]. The implementation of environmental responsibility relies on the joint participation of different environmental stakeholders, including consumers, manufacturers, the public, governments, and businesses [37,38,39]. Therefore, stakeholder involvement is considered a key driver of environmental responsibility [40,41,42].

The online channel provides a new window of development for the service sector. China’s e-tailing, online take-out, and new food e-commerce user scale reached 749 million, 409 million, and 257 million users, respectively, with the e-tailing market remaining the world’s number one for seven consecutive years. In recent years, the take-out industry has proliferated, and the proliferation of take-out packaging has worsened the problem of plastic pollution. While providing convenience, it has also generated environmental problems [43]. The development of the new economy has led to excessive packaging waste from take-out, plastic waste from the catering industry, and food waste, among other environmental issues, which have become an important new area of focus.

The catering industry’s environmental problems mainly include wastewater and air emissions, kitchen waste disposal, and plastic take-out packaging. National provinces and municipalities are governed by corresponding laws and regulations, and enterprises must meet emission standards to reduce environmental pollution. The government has also restricted the management of plastic packaging and plastic straws for dine-in and take-out. However, with the rapid development of the take-out industry, many orders are placed that compound waste resources and pollute the environment; the extent of the problem cannot be ignored, and it has become a fundamental social challenge for the new era. This kind of environmental pollution cannot be solved by enterprises alone; it also requires the participation of consumers. The problem of plastic waste catering requires input from both upstream through technical product innovation and downstream through consumer demand.

### 2.2. Managing CSR through Customer Participation

CSR is considered an important strategic tool for companies to actively solve social problems [44,45]. CSR is not only a legal responsibility for a company but also a voluntary pro-social activity for the enterprise to achieve sustainable development. CSR can improve financial performance [21], attract investment and employment [41,46], build corporate image [47], and play an insurance role [48]. By striving to create an image of responsible corporate citizenship, companies will gain a good reputation, brand value, and consumer trust and increase their market value. Modern companies are increasingly proactive in adopting proactive CSR strategies and promoting sustainability activities to address current environmental and social challenges [17,49].

Customer engagement includes emotional contributions to a company’s service process beyond information, physical, behavioral, and consumption features [50,51]. Consumer participation behaviors include not only information seeking, information sharing, and responsible behavior but also interactions between consumers and brands. In addition, the increasing use of online social media has made it easier for customers to interact with brands or other customers [52]. Consumer CSR participation is critical to a company’s CSR [18,53]. Research suggests that customer engagement behavior is heavily influenced by how they perceive a company’s CSR behavior [54,55]. Customers’ CSR perceptions influence consumers’ purchase, loyalty, and recommendation intentions toward brands, products, and services.

There are two important roles in signaling theory, signaler and receiver. Specifically, signalers send signals to receivers to reduce information asymmetry, while receivers need to respond to signals and provide feedback to signalers [56,57]. The CSR strategy of enterprises sends a positive signal to consumers that they are socially responsible corporate citizens [58,59], and consumer engagement is a response behavior as a receiver.

CSR participation involves consumers volunteering their time, effort, and money to participate in brand social responsibility. CSR participation includes purchases based on good cause marketing, participation in brand social responsibility activities, and supporting and retweeting brand initiatives through social media (e.g., likes, comments, and retweets). Customers are increasingly engaged in brand social responsibility, such as Saturnbird Coffee’s return program (This return program is a long-term plan to recycle empty coffee shells. The empty shells are exchanged for themed materials, and the recycled packaging shells are reused to make other products), Alipay Ant Forest (Alipay users who engage in low-carbon acts, like renting a bike, taking the bus, or simply walking, are rewarded with “green energy,” which is used to “water” virtual trees in their mobile phones. When a virtual tree grows up, a real tree is planted by Ant Forest), and Coca-Cola’s 2ndlives activity (To encourage people to recycle and reuse, Coca-Cola, together with Ogilvy & Mather China, launched a campaign called “2ndlives” in Thailand and Vietnam, in which Coca-Cola provided people with 16 free functional bottle caps that could be screwed onto old Coke bottles to turn them into various household utensils). Co-creating value with customers is one of the most powerful forces driving future business growth [60,61]. Customer involvement in CSR can reduce consumer suspicion [36], increase the efficiency of CSR activities [62], enhance customer–organization relationships [63], and improve loyalty and brand reputation [53]. Li et al. find that corporate environmental responsibility engagement affects corporate value [64]. This study focuses on consumers’ willingness to participate in CSR initiatives to improve the efficiency of managing BER.

### 2.3. CSR Communication and Transparency Signaling on Social Media

Social media and online platforms have changed the way people interact with each other [65]. The digital age is changing the way companies communicate with consumers and innovating socially responsible practices and communication models [18,66]. Digital technologies provide companies with the opportunity for complete transparency in work processes and value chains [67]. Digital technologies allow for online tracking of each product’s production sources, working conditions, and environmental footprint. Social media has become an essential part of everyday life and has improved people’s preferences and needs based on the information they share on social networks [19,68]. Companies use online social media to share information about their products, receive feedback from consumers, make announcements about company activities, and engage with communities and the public [69,70].

In a CSR context, Kim et al. introduce three communication strategies with different stakeholder participation levels: information, response, and involvement strategies [17]. Heikkurinen and Forsman-Hugg classify reactive and proactive CSR as responsive strategic corporate responsibility by exploring the relationship between the responsibilities of the Finnish food supply chain and corporate strategy [71]. Proactive CSR aims to enhance competitive advantage, while reactive CSR aims to maintain it. Proactive CSR has a forward-looking and altruistic attribute. Companies proactively undertake socially responsible activities without environmental pressure or adverse reports [72]. In contrast, reactive CSR is a defensive measure taken by companies to protect a company’s reputation and brand image and is reactive and egoistic [73].

CSR communication through social media can improve information transparency and increase interactions with consumers. Scholars have defined the concept of transparency in different ways. Rim et al. define transparency as “including good and bad CSR information disclosure [74].” From the firm’s level, transparency is defined as the magnitude of visibility and accessibility of information [75]. From the consumer’s level, transparency is a subjective judgment of the extent to which information is held about a company in an interaction [76]. Transparency is morally important because it shows honesty, openness, and commitment to the truth.

Transparency signals have been recognized as a critical element in CSR communication to help companies build trust-based relationships with the public [21,77]. Corporate transparency moderates the relationship between CSR and brand attachment [78]. Transparency in corporate communication leads to more positive attitudes toward environmental product claims than a lack of transparency [24], feeling better about making socially/environmentally conscious purchases, and a willingness to pay higher prices [79,80]. While existing research suggests that transparency signals lead to positive consumer responses, little is known about how to use CSR transparency signals (high vs. low) in consumer decisions. In addition, prior research has not distinguished between different CSR strategies (proactive vs. reactive). Based on the literature, this study expects that transparency signals reduce information asymmetry and serve a role in resolving dissonance under social media communication or enhancing active CSR information stimulus participation, even though consumers make brand-related decisions based on their own perceptions. Moreover, the signal effects of CSR strategy may vary with the level of transparency signal according to the signaling theory [13,14]; that is, different levels of transparency may change the effectiveness of CSR. Therefore, the following hypotheses are proposed.

**H1a:** 
*A proactive CSR strategy has a more significant positive effect on BER participation intention when transparency is high (vs. low).*


**H1b:** 
*A reactive brand social responsibility strategy has a more significant positive effect on BER participation intention when transparency is low (vs. high).*


### 2.4. Customer Trust

Efficient and transparent social responsibility communication can improve consumer trust [81]. Trust is defined as the “expectation of ethical conduct” and is considered “the confidence and will of one party to the other” [82]. Trust is customers’ expectations and perceptions of socially responsible or ethically justified corporate behavior from a CSR perspective. Previous research has divided trust into two subdimensions: integrity and competence. Integrity is described as the belief that an organization is fair and impartial, while competence is the belief that an organization can do what it says it wants to do. Competence-based trust relies on an organization having a wealth of knowledge and experience; integrity-based trust depends on the organization being honest, open, and concerned about the public interest.

Several scholars have examined the effects of competence-based and integrity-based trust. Integrity-based trust has a more profound impact on public acceptance of renewable energy projects than competence-based trust [82]. Connelly et al. find that integrity-based trust is approximately 10 times more efficient at reducing transaction costs in inter-organizational relationships than competence-based trust [83]. Perceived competence has a greater effect on buyer purchases than perceived integrity [84]. Terwel et al. validate the attitudes of competence-based trust and integrity-based trust toward public participation in the management and use of new technologies, respectively [85]. Previous studies find a difference in the effects of the two trust types.

We define a transparency signal as a definitive piece of information that is accessible, available, and characterized by objectivity and truthfulness [78]. In short, companies and brands should provide clear, understandable, objective, and truthful information to third parties [25]. Thus, both CSR and transparency can be considered as a source of signals [26]. Consumers can judge the moral quality of a company based on a combination of transparency signals and CSR information cues. Therefore, companies can reconcile the two signal cues to mitigate the adverse consequences of information asymmetry for consumers. High transparency signals are more likely to provide consumers with clearer and more valuable information about a company or brand, increasing trust and reducing uncertainty in communication [86]. In our study, we hypothesize the following effect of the interaction between CSR strategy and transparency signals on the two dimensions of trust.

**H2a:** 
*A proactive CSR strategy has a more significant positive effect on consumers’ perceived integrity when transparency is high (vs. low).*


**H2b:** 
*A reactive brand social responsibility strategy has a more significant positive effect on consumers’ perceived integrity when transparency is low (vs. high).*


**H3a:** 
*A proactive CSR strategy has a greater positive effect on consumers’ perceived competence when transparency is high (vs. low).*


**H3b:** 
*A reactive brand social responsibility strategy has a greater positive effect on consumers’ perceived competence when transparency is low (vs. high).*


### 2.5. Customer–Brand Identification

Brand identity is often defined as “the consumer’s personal subjective feelings about the ownership and identity of the brand” [87,88]. The brand’s CSR activities may increase the company’s focus on environmental, social, and philanthropic aspects [89]. It may simultaneously make consumers think that it is a good brand worthy of admiration and trust [90]. Therefore, consumers may identify with a brand and be willing to buy its products and services if they feel admiration and warmth [91]. Consumers feel trust in the brand and consider it trustworthy [72]. Brand attachment and brand advocacy are formed between consumers and the brand [27,92].

Based on psychological distance theory, identification reduces psychological distance and thus increases willingness to participate owing to identification with the brand’s social responsibility activities. Consumers’ positive participation in CSR activities is one of the most potent forms of support for CSR. Purchasing products or services is another form of participation in CSR activities. Previous research [93] has found that, based on social identity theory and source credibility theory, it explores how customers’ CSR perception influences their willingness to participate, with customer–company identification being a vital mediator moderated by CSR credibility and company trust. Lee et al. investigated consumers’ use of CSR communication channels and the mechanisms by which consumers’ CSR awareness led them to engage in CSR activities [94].

According to social identity theory [28,29], people identify with a brand when they believe it can maintain and enhance their self-esteem. Because consumers identify with the brand, participating in the brand’s journey of “doing good” can be a way to demonstrate the customer’s ego, enhancing their self-image. At this time, customers and brands have consistent cognition and values. Thus, we propose the following hypotheses.

**H4a:** 
*A proactive CSR strategy has a greater positive effect on CBI when transparency is high (vs. low).*


**H4b:** 
*A reactive brand social responsibility strategy has a greater positive effect on CBI when transparency is low (vs. high).*


**H5:** 
*The impact of the interaction between BER strategy and transparency signaling on BER participation intention is serially mediated by perceived integrity and CBI.*


**H6:** 
*The impact of the interaction between BER strategy and transparency signaling on BER participation intention is serially mediated by perceived competence and CBI.*


Figure 1 summarizes our proposed research hypotheses.

## 3. Methodology

### 3.1. Stimulus Development and Pilot Test

We conducted a pilot test that used reading tasks to confirm the effectiveness of the CSR strategy and transparency signal (i.e., four sets of stimulus materials) manipulation (see Appendix A). Stimulus materials were distributed to students at a university via an online questionnaire platform (Sojump), and 80 participants took part in the pilot study. Respondents were asked to randomly read one of four sets of informational materials and answer manipulation check questions. The results of the pilot tests were used to adjust and revise the experimental stimulus materials by (a) shortening the CSR strategic information statement, (b) modifying the transparency manipulation information of brand responsibility activities, and (c) reformulating the statement used to measure the research structure.

Following prior research using CSR strategy messages, we developed two CSR strategy messages for the study. To check the manipulation of the nature of CSR strategy, two 7-point bipolar items measured respondents’ level of agreement or disagreement with the CSR message: reactive–proactive and involuntary–voluntary. For the two-level manipulation transparency, we prepared BER activity information stimulus materials for two scenarios. We used four items to measure transparency signals [78] (see Appendix B). The stimulus material is adapted from the CSR information on the official McDonald’s China website.

The results of the *t*-test of independent samples showed that respondents who received information on active CSR strategies reported higher levels of corporate initiative than those who received information on passive CSR strategies (M _proactive_ = 5.26, M _reactive_ = 4.22; *t* (78) = 4.07, *p* < 0.01). Concerning the manipulation of information transparency, the results show that our manipulation of transparency was successful (M _high_ = 5.10, M _low_ = 3.86; *t* (68.43) = 4.29, *p* < 0.01).

### 3.2. Procedures and Instruments

The study used virtual brand names to reduce the impact of previous experiences and brand factors on consumers. Four sets of stimulation materials were designed to manipulate the nature of CSR strategy and the level of CSR information transparency. Questionnaire links were pushed to individuals by WeChat, and one of the four CSR scenarios was randomly delivered to each participant. The questionnaire was divided into three parts. Part I included participants’ demographic information. Part II contained the study instructions and the CSR information stimulation materials. After reviewing the situation (manipulation check question), participants were asked to evaluate questions in Part III, which were related to their perceived brand integrity, perceived brand competence, CBI, and willingness to participate in BER activities, according to the subjective perception of the subjects.

Four items were developed based on Lee et al. and Beldad et al. to measure customer BER participation intention (α = 0.924) [62,94]. Adapting Mael and Ashforth, and Hur et al. [42,95], we used four items to measure CBI (α = 0.922). Perceived integrity (α = 0.908) was measured using four items from Rim et al. [74] and Cambier and Poncin [77]. Three items were used to measure customers’ perceived corporate competence (α = 0.895) by Aaker et al. [96] and Rim et al. [74].

To represent observable constructs for each latent construct, all measurements used a 7-point Likert scale. The brand environmental participation intention used the formulation 1 = completely impossible, 7 = completely possible, and the remaining measurements used the formulation 1 = strongly disagree, 7 = strongly agree. The reliability and validity indexes of the variables are shown in Table 1 and Table 2, respectively. The Cronbach’s alpha and CR values of each variable were above 0.8, demonstrating that the measures had sufficiently high internal consistency.

### 3.3. Data Collection and Analysis Methods

The public opinion survey platform in China (www.wjx.cn) offers functions equivalent to Amazon Mechanical Turk. Ultimately, 140 effective responses were obtained. Table 3 displays the means and standard deviations for individual experimental groups and for individual dependent variables. To test the proposed hypotheses, a two-way multivariate analysis of covariance (MANCOVA) was conducted using SPSS 26. Perceived integrity, perceived competence, CBI, and BER participation intention were the dependent variables. In addition, PROCESS Model 86 was used for testing serial mediation analysis [97,98]. The sample size was set to 5000, and the confidence interval was set to 95%.

## 4. Results

### 4.1. Respondents’ Characteristics

The survey was conducted in May 2021 through an online professional questionnaire platform. A total of 153 questionnaires were collected from Chinese consumers. After rigorous checking by two doctoral students, 13 questionnaires with incomplete information, inconsistent text, or the same IP address were excluded. The final 140 effective questionnaires (91.5% effective return rate) were collected for further data analysis. The demographic information is shown in Table 4. Among the respondents, 54% were male, and 46% were female. Regarding age, 16% were 18–25 years, 22% were 26–30 years, 25% were 31–40 years, 20% were 41–50 years, and 13% were over 50 years.

### 4.2. Manipulation Check

The *t*-test results for independent samples indicated that these operations were considered valid. Participants who read proactive CSR statements report more voluntary brand involvement in CSR (M _proactive_ = 5.53) than participants who read reactive CSR statements (M _reactive_ = 2.97; *t* (140) = 16.544, *p* < 0.001). Manipulation checks for transparency signals were significant, such that the high-transparency situations scored higher (M _high_ = 5.66) than the low-transparency situations (M _low_ = 3.60; *t* (140) = 14.116, *p* < 0.001). Therefore, the manipulation test of this study was successful.

### 4.3. Hypothesis Test

#### 4.3.1. Two-Way Interaction Effects between CSR Strategy and Transparency Signaling

To test the combined effect of CSR strategy and transparency signals, a two-way MANCOVA was conducted. CSR strategy (proactive/reactive) and transparency signals (high/low) were the independent variables, and perceived integrity, perceived competence, and intention to participate in BER were the dependent variables.

The hypothesis of this study is that there would be different effects on perceived integrity (H2) and perceived competence (H3), CBI (H4), and intention to engage in BER practices (H1) at different levels of transparency signals and with different nature of CSR strategies. The effects of CSR strategy and transparency signal on perceived integrity, perceived competence, CBI, and participation intention toward BER were nonsignificant (*p* > 0.05). However, the MANCOVA results showed that the combined effect of CSR strategy and transparency signal was significant (Wilks’ lambda = 0.93, F (3, 206) = 5.38, *p* < 0.01). Table 5 presents the MANCOVA results.

The MANCOVA results indicated that the interaction effect of CSR strategy and transparency signal on perceived integrity was significant (F = 0.753, *p* < 0.05; Table 5). According to the results, when exposed to proactive CSR strategy messages, respondents who read high transparency signal initiatives reported a higher level of perceived integrity (M _proactive-high_ = 5.55, SD = 0.88) than those who read low transparency signal initiatives (M _proactive-low_ = 4.26, SD = 1.32). Conversely, when exposed to reactive CSR strategy messages, respondents who received high transparency signal initiatives reported a higher level of perceived integrity (M _reactive-high_ = 5.63, SD = 1.03) than those who received low transparency signal initiatives (M _reactive-low_ = 4.03, SD = 1.12). Therefore, H2 was supported.

We conducted two-way ANOVAs on perceived competence to test H3. The analyses revealed a significant interaction effect of CSR strategy and transparency signals on perceived competence (F = 0.39, *p* < 0.05). When exposed to proactive CSR strategy messages, respondents who read high transparency signal initiatives reported higher levels of perceived competence (M _proactive-high_ = 5.48, SD = 1.23) than those who read low transparency signal initiatives (M _proactive-low_ = 4.73, SD = 1.43). However, when exposed to reactive CSR strategy messages, respondents who viewed high transparency information reported higher levels of perceived competence (M _reactive-high_ = 4.95, SD = 1.46) than those who received low transparency information (M _reactive-low_ = 4.49, SD = 1.27). Thus, H3 was supported.

Furthermore, ANOVA results indicated that there was no two-way interaction effect between the nature of CSR strategy and transparency signals on CBI (F = 0.002, *p* > 0.05). The results of an independent sample *t*-test demonstrate that consumers reported better CBI toward proactive CSR strategy messages when they received a high transparency signal than a low transparency signal (M _proactive-high_ = 5.23, SD = 0.83 vs. M _proactive-low_ = 4.87). Similarly, when exposed to reactive CSR strategy messages, respondents who received high transparency information reported higher levels of CBI (M _reactive-high_ = 4.85, SD = 1.49) than those who received low transparency signal information (M _reactive-low_ = 4.51, SD = 1.45). These results reject H4.

Finally, we examined the interaction effects between the nature of CSR strategy and the level of transparency signal on customer BER participation intention. The interaction was significant (F = 0.138, *p* < 0.001). When exposed to proactive CSR strategy messages, respondents who received high transparency signal initiatives reported a higher level of BER participation intentions (M _proactive-high_ = 5.65, SD = 0.73) than those who received low transparency signal initiatives (M _proactive-low_ = 4.64, SD = 1.62). Conversely, when exposed to reactive CSR strategy messages, respondents who received low transparency signal initiatives reported a higher level of BEC participation intentions (M _reactive-low_ = 3.68, SD = 1.5) than those who received high transparency signal initiatives (M _reactive- high_ = 3.53, SD = 1.42). Therefore, H1 was supported.

#### 4.3.2. Serial Mediation Analysis

This study hypothesized that the interaction effect between CSR strategy and transparency signaling on BER participation intention is continuously mediated by perceived integrity and CBI. We use the bootstrap method by process model 86 for mediation analysis to test this hypothesis [97]. In this study, BER participation intention is used as the dependent variable, CSR strategy as the independent variable, transparency signal as a moderator, and perceived integrity and CBI as mediators (Figure 2).

The bootstrapping results indicated a significant “CSR strategy→perceived integrity→CBI→BER participation intention” serial mediation process in the high transparency signal (indirect effect = 0.08; 95% CI = 0.01 to 0.19) but not in the low transparency signal (indirect effect = −0.02; 95% CI = −0.107 to 0.147, n.s.). The indirect effect of “CSR strategy→perceived integrity→BER participation intention” was not significant in the high transparency signal (indirect effect = −0.09; 95% CI = −0.276 to 0.03, n.s.) or in the low transparency signal (indirect effect = 0.01; 95% CI = −0.07 to 0.09, n.s.). The mediation effect results show that perceived integrity is not significant when it is used as a single mediator, but it is significant through serial mediation of perceived integrity and CBI. In summary, these results support H5.

Similarly, we tested our full conceptual model (CSR strategy×transparency signaling→perceived competence→CBI→BER participation intention), as shown in Figure 3. The analyses showed a significant moderated serial mediation effect on BER participation intention. We further probed the conditional indirect effects and found significant indirect effects serially through perceived competence and CBI only when the transparency signal was high (indirect effect = 0.18; 95% CI = 0.05 to 0.34). Therefore, H6 was supported.

## 5. Discussion and Conclusion

### 5.1. Major Findings and Implications

By integrating signaling theory and social identity theory, we consider that enterprises use CSR strategy (proactive vs. reactive) as a signal to strengthen consumer trust (integrity and competence) and customer–brand identification so as to promote customer BER participation intentions and different levels of transparency signals will affect the signal effect of CSR strategy. The corresponding research results contribute to the literature on CSR and brand management in several ways. First, we extend previous research by examining consumer BER participation intention under the interaction effect between the different natures of CSR strategies and different levels of transparency signals. Second, we consider perceived integrity, competence, and CBI as potential psychological mechanisms to explain customers’ willingness to engage in brand social responsibility. Third, we shed light on the persuasive mechanisms of brand social responsibility communication by identifying three key mediators—perceived integrity, perceived competence, and CBI—and by testing two moderated mediation models in the context of CSR communication under social media.

### 5.2. Theoretical Contributions

The current study makes several theoretical contributions to the literature. First, it contributes to recent research on transparency signals. Due to the widespread use of information technology and social media, consumers are demanding greater information transparency from brands and companies [78,99]. This study finds that transparency is critical to direct stakeholders, expressing corporate honesty and integrity, and accurately communicating the brand’s social responsibility values to consumers. Transparency is, therefore, a boundary condition for the effectiveness of CSR practices. CSR strategy and transparency need to match each other to improve consumer perception and engagement. We connect CSR strategies with transparency signals to enrich the knowledge of how effective CSR implementation works and when it benefits the company. Although previous studies have confirmed that transparency is a prerequisite for CSR reporting [22,23,100], the boundary conditions of transparency on CSR effectiveness have not been explored. Our findings have enriched the comprehension and application of transparency signals in CSR management. We find that transparency is beneficial for firms in terms of improving CSR effectiveness, which increases consumers’ willingness to engage with BER in the case of proactive high-transparency information. This research demonstrates the differential effects of CSR strategy (proactive or reactive) on persuasive messages with various transparency signals (high or low). Previous studies have investigated the signaling effect of brand transparency on perceived brand integrity in marketing communications [77]. Heinberg et al. investigate different types of corporate transparency as a boundary condition of the effects of CSR activities on the consumer–brand relationship [78]. Our findings suggest that the match between a proactive (reactive) CSR strategy and a high (low) transparency signal can effectively encourage consumers’ BER participation intention and behaviors.

Second, we investigated the effectiveness of CSR practices and management from the perspective of consumers’ information processing and psychological mechanisms. Consumers are potentially overwhelmed by the vast amount of information accessible in the Internet age. As a result, they have higher standards for information, both in terms of the content provided and the media channels offered by the brand. Therefore, when making decisions, consumers rely on specific brand signals, such as the nature of CSR and transparency. In the case of conflicting signals, consumers try to uncover the underlying motives of the brand. This process may be particularly detrimental to CSR-related signals, as doubts about the motivation to do good are a key reason for poor CSR effectiveness [36,101]. Prior research has shown that if a company performs well in both the CSR and corporate transparency domains, consumer skepticism is dampened, thereby strengthening the link between CSR and brand attachment [102]. Scholars and practitioners have emphasized the importance of corporate communication transparency, as it mitigates public skepticism [24].

Finally, this research extends prior studies by using CBI as a key serial mediator to ascertain the psychological mechanisms of how customers’ perceptions lead to their BER participation intentions. In other words, consumers’ awareness and understanding of corporate image, corporate history, and corporate values influence consumer engagement with CSR and BER. Our findings are consistent with previous studies in which customers’ perspectives on historical CSR affected their intention to participate [93,103,104]. Our results also support the findings of Einwiller et al. and W.-M. Hur et al., who show that customer corporate identification mediates the relationship between customers’ CSR perceptions and their donations to nonprofit organizations affiliated with the company [93,105]. The results of our study support social identity theory by explaining the mediating effects of C–B identification in the relationship between customers’ perceptions and their BER participation.

### 5.3. Practical Implications

This study provides practical implications for CSR professionals in the following areas. First, business managers need to develop clear strategies and goals for social responsibility in their companies, such as whether a company’s focus is on responding to stakeholder pressure to prevent environmental problems or on proactively creating better environmental quality, or whether it requires customer participation in corporate environmental initiatives to co-create value [61].

Second, after establishing the corresponding CSR strategy, brand communication managers need to pay attention to transparency signaling since different transparency signals can have different effects on customer behavior under different social responsibility strategies [106]. According to our findings, the match between proactive (high transparency) and reactive (low transparency) can lead to more positive consumer perceptions of perceived trust, customer brand identity, and participation in green initiatives. When companies use proactive CSR strategies, high transparency information (e.g., ability to see other people’s comments, comparative information with competitors, greater accessibility, and better understanding) can enhance customer trust (perceived integrity and perceived competence), CBI, and green participation. When companies use reactive CSR strategies, low-transparency information is more likely to incentivize consumer participation behavior. CSR practices require investment in time, money, and human resources, and most well-known companies with long-term and social responsibility values adopt proactive strategies [107]. However, small and medium-sized enterprises are usually in survival mode. They need to spend more resources on business development, and thus, they are more likely to choose a reactive social responsibility strategy [108]. In the era of digital transparency, different CSR strategies must match the corresponding transparency signals in brand communication.

Finally, to encourage customer participation to incorporate environmental initiatives in the future, companies need to increase customer trust and recognition of their brands. This would make them more likely to feel good about CSR communication and, thus, participate in related environmental initiatives [24]. Helping customers perceive a company’s integrity and competence and helping them build brand identity are critical to customer participation in CSR activities [85]. When customers develop trust in CSR communications that the company is honest and competent, it can lead to customer engagement. Brand identity can mediate the impact of CSR communication on customer engagement.

### 5.4. Limitations and Suggestions for Future Research

This study has several limitations that should be addressed in future research. First, while a fictitious company can minimize the adverse effects of an individual’s prior experience with a known brand, future research should replicate this study using an authentic brand to increase the study’s external validity. This study measures behavioral intentions rather than actual behavior. Future research should investigate actual engagement behaviors.

Second, the study manipulates only two levels of transparency: high and low. A high level of transparency is signaled by the accessibility and objectivity of CSR information. Although the manipulation of information transparency in this study is based on existing literature, considering different industry contexts, CSR themes, and definitions of transparency would yield different manipulation results.

Third, we use the interaction of CSR strategic attributes and information transparency as independent variables but do not consider customer characteristics, such as gender, age, construal level, and cultural orientation (i.e., collectivism). Previous studies [103,109] have analyzed how these variables moderate customers’ CSR perceptions and their effects. Therefore, we propose elaborating on the relationship between customer perceptions of CSR and CSR participation using consumer characteristics as moderating factors. We choose only perceived honesty, perceived competence, and consumer brand identity as variables for evaluating consumers’ internal states; other psychological variables could be explored in the future.

## Figures and Tables

**Figure 1 behavsci-12-00514-f001:**
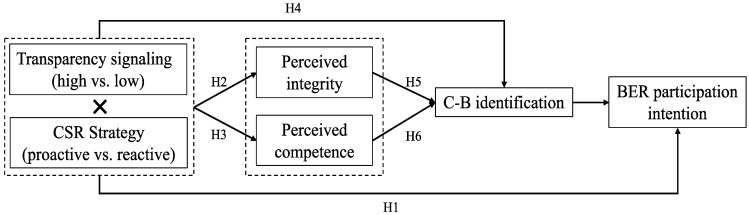
Proposed research model.

**Figure 2 behavsci-12-00514-f002:**
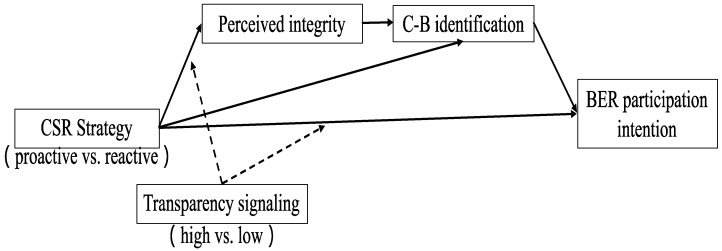
The “CSR strategy × Transparency signaling→perceived intergrity→CBI→BER participation intention” moderated serial mediation.

**Figure 3 behavsci-12-00514-f003:**
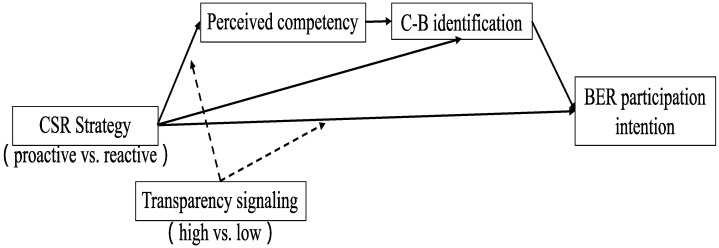
The “CSR strategy × Transparency signaling → perceived competence → CBI→ BER participation intention” moderated serial mediation.

**Table 1 behavsci-12-00514-t001:** Reliability of Constructs.

Measures and Items (7-Point Likert Scales)	Factor Loading	Cronbach’s α	CR	AVE
Perceived integrity (PI)				
- This brand treats people like me fairly and justly	0.802 ***	0.908	0.911	0.72
- Whenever this brand makes an important decision, I know it will be concerned about people like me	0.886 ***			
- Sound principles seem to guide this brand’s behavior	0.892 ***			
- This brand does not mislead people like me	0.804 ***			
Perceived competence (PC)				
- I feel very confident about this brand’s skills	0.854 ***	0.895	0.899	0.748
- This brand can accomplish what it says it will do	0.919 ***			
- This brand is known to be successful at the things it tries to do	0.817 ***			
Customer-Brand identification (CBI)				
- When someone criticizes this brand, it feels like a personal insult.	0.788 ***	0.922	0.926	0.759
- When I talk about this brand, I usually say “we” rather than “they.”	0.905 ***			
- This brand’s successes are my successes.	0.928 ***			
- When someone praises this brand, it feels like a personal compliment.	0.849 ***			
BER participation intention (BPI)				
- It’s probable that I will be involved in the brand’s environmental services programs	0.846 ***	0.924	0.924	0.753
- My involvement in the brand’s environmental services programs is likely	0.837 ***			
- I am willing to get involved in the brand’s environmental services programs	0.913 ***			
- I would consider getting involved in the brand’s environmental services programs	0.873 ***			

Note: *** *p* < 0.001.

**Table 2 behavsci-12-00514-t002:** Correlations of the constructs.

	PI	PC	CBI	BPI
Perceived integrity (PI)	0.849			
Perceived competence (PC)	0.597 **	0.865		
Customer-Band identification (CBI)	0.653 **	0.691 **	0.871	
BER participation intention (BPI)	0.273 **	0.298 **	0.205 *	0.868

Note: Diagonal elements are the square root of the AVE. Off-diagonal elements are the correlations among the constructs. * *p* < 0.05, ** *p* < 0.01.

**Table 3 behavsci-12-00514-t003:** Means and standard deviations for dependent variables.

CSR Strategy	Transparency Signal	Perceived Integrity		Perceived Competence		CBI		BER Participation Intention	
		M	SD	M	SD	M	SD	M	SD
Proactive	High	5.55	0.88	5.48	1.23	5.23	0.83	5.65	0.73
	Low	4.26	1.32	4.73	1.43	4.87	1.6	4.64	1.62
Reactive	High	5.63	1.03	4.95	1.46	4.85	1.49	3.53	1.42
	Low	4.03	1.12	4.49	1.27	4.51	1.45	3.68	1.5

Note: Customer-Band identification (CBI), Brand environment responsibility (BER).

**Table 4 behavsci-12-00514-t004:** Descriptive profile of the respondents.

	N	%
Gender		
Male	75	53.57%
Female	65	46.43%
Age		
Younger than age 18 years	5	3.57%
18~25 years	22	15.71%
26~30 years	31	22.14%
31~40 years	35	25.00%
41~50 years	29	20.71%
51~60 years	14	10.00%
Over than age of 60 years	4	2.86%

**Table 5 behavsci-12-00514-t005:** Analysis of covariance results for dependent variables (F-values).

	Perceived Integrity	Perceived Competence	CBI	BER Participation Intention
Main effect				
CSR Strategy (CS)	0.164 **	2.78 **	1742.48 **	28.32 **
Transparency signaling (TS)	60.375 ***	6.89	2.49	10.36
Interaction effect				
CS * TS	0.753 *	0.39 *	0.002	0.138 **
Verification	H2 Accepted	H3 Accepted	H4 Rejected	H1 Accepted

Note: * *p* < 0.05, ** *p* < 0.01, *** *p* < 0.001; df (1, 165).

## Data Availability

The dataset of this study is available from the corresponding author on reasonable request.

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
