# Peer review of "Does Brand Truth-Telling Yield Customer Participation? The Interaction Effects of CSR Strategy and Transparency Signaling"

_behavsci, 2022, doi:10.3390/bs12120514_

Round 1
Reviewer 1 Report
Thank you for the submission. This is a good paper that covers an interesting topic. There are a few issues with the manuscript that the authors will do well to address as they continue working on improving the paper.
1. Introduction: Though the Introduction section explained why the topic is interesting and what key empirical findings have already informed the topic in question, it lacks a good justification and literature support on why the study is needed. Meanwhile, why applying signaling theory and social identity theory in the study was not well justified. In other words, what is the research gap? What are the research questions answering in the study?
2. More literature needs to support signaling theory and social identity theory.
3. Methodology: Any modifications from the results of pilot study? Can't find any Appendix attached. Who were the participants? Any screening questions? What's the sample size? How's it determined?
4. Results: It should be cautious when include participants who are younger than 18 years old. It is hard to evaluate without seeing the Appendix (manipulation).
5. Implications for CSR strategies should be specific and exemplified. Such as, any examples for CSR communications?
Reviewer 2 Report
I read the study closely: excellent, well-written, interesting. The author's conclusions make common sense to me. I suggest a revised title:
Brand Truth-Telling and Customer Participation. CSR Strategy and Transparency in China’s E-tailing Sector a Case Study.

Reviewer 3 Report
The topic investigated is very interesting and presents significant theoretical and empirical contributions.
However, it could be improved in some aspects.
- Introduction- Deepen the pertinence of developing this investigation;
- Review the presentation of abbreviations to facilitate a better understanding of the text (CSR; BER; CBI; BSR);
- The methodology used is adequate, as well as the statistical treatment; nevertheless the results may be presented with more clarity.
For example, the bootstrapping technique was used, what is the number of reference samples?.
For example, tables showing the results of the effects, coefficient of determination could be presented....
For example, figure 5 could also be enriched in order to allow a quick reading of the results, that is, to visualize whether or not the hypothesis was rejected.
I also consider the limitations and suggestions for future research appropriate.
Reviewer 4 Report
Congratulations for interesting article!
One thing: making conclusions from 153 respondents out of "China consumers" is a big too optimistic (quantitative research), however the overall structure of applied methodology is sound.
Round 2
Reviewer 1 Report
Thanks for revising the manuscript. Authors have addressed most of my concerns. Additional comments are listed below.
1. In Appendix B, how the BER activity information stimulus materials for two scenarios were developed? The only difference between High Transparency and Low Transparency is a paragraph of words "According to consumer feedback...". I'm just wondering how the validity can be ensured and verified.
2. During the pretest, what modifications were done to adjust the linguistic expression of the stimulus material in Appendix A? They need to be explained in the manuscript.
